# Motivation to access laparoscopic skills training: Results of a Canadian survey of obstetrics and gynecology residents

Jocelyn Stairs[1]*, Bradley W. Bergey[2], Finlay Maguire[3], Stephanie Scott[4]

**1** Department of Obstetrics and Gynecology, Dalhousie University, Halifax, Nova Scotia, Canada, **2** Division of Education, Queens College, City University of New York, New York, New York, United States of America, **3** Faculty of Computer Science, Dalhousie University, Halifax, Nova Scotia, Canada, **4** Division of Gynecologic-Oncology, Dalhousie University, Halifax, Nova Scotia, Canada

* jocelyn.stairs@dal.ca

## Abstract

### Objective

Competency based medical education (CBME) requires novel approaches to surgical education. Significant investment has been made in laparoscopic simulation, which has been shown to foster skill development prior to patient encounters. However, research suggests variable voluntary use of these resources by residents, and little is known about the motivational factors that influence their utilization. The purpose of this study was to characterize factors that motivate residents to seek laparoscopic simulation experience outside of the formal curriculum.

### Design

We developed a questionnaire grounded in Expectancy Value Theory, an established psychological theory of motivation, by adapting validated measures to fit the study context. We conducted a cross sectional survey of Canadian obstetrics and gynecology residents.

### Setting

We invited residents enrolled in English-language obstetrics and gynecology training programs in Canada to participate.

### Participants

All residents engaged in clinical duties during Winter 2018 were invited to complete the questionnaire. Forty-four Obstetrics and Gynecology (Ob/Gyn) residents participated in the study.

### Results

Residents reported limited use of simulation resources and identified multiple barriers including lack of time, access, and supervision. They expressed concern about development of bad habits during independent practice, and simulation use was positively

**Data Availability Statement:** The small number of residents in each year of training and the small number of Canadian training sites makes the potential for identification possible. Given the

potential implications of residents critiquing their own institution's teaching environment, even if they were only identified in aggregate by site, our Institutional Ethics Review Board considered the raw responses as sensitive. They can be contacted here: IWK Health Centre Research Services 5850/5980 University Ave PO Box 9700 Halifax, Nova Scotia, Canada B3K 6R8 research@iwk.nshealth.ca

**Funding:** JS was awarded the Dalhousie University Department of Obstetrics and Gynaecology HB Atlee Endowment Fund Award 2018 These funds come from an endowment that was bequeathed by Harold Benge Atlee upon his death. The funders played no role in any part of study design, data collection, decision to publish, or manuscript preparation. Details of the award can be found here: https://cdn.dal.ca/content/dam/dalhousie/pdf/faculty/medicine/departments/department-sites/obstetrics/TOR-Atlee-Award2012.pdf

**Competing interests:** The authors have declared that no competing interests exist.

correlated with perceived utility. Compared to junior residents, senior residents reported greater enjoyment of laparoscopic surgery, less emotional costs, and higher self-efficacy for learning laparoscopy.

## Conclusions

Residents' perception of utility and barriers impede voluntary simulation use and overall use was limited. As programs undertake curricula redevelopment for CBME, mitigating barriers and improving perceived utility of laparoscopic simulation is vital to increase use and enhance skill development.

## Introduction

Obstetrics and Gynecology residency programs in Canada are shifting to a competency-based medical education model under the guidance of the Royal College of Physicians and Surgeons of Canada [1]. This outcome-based rather than time-based model of training emphasizes increased learner responsibility for skill acquisition [1]. Laparoscopy is a fundamental surgical skill to the practice of gynecology, and resident skill development in this area is central to the development of competence in gynecologic surgery. With heightened attention to patient safety, operating room time contraints, increased resident cohort size, and resident work hour restrictions, simulation is an increasingly important mode of resident laparoscopic skill acquisition [2–6].

Simulation has been shown to facilitate laparoscopic skill acquisition and be transferrable to the operating room [2, 7, 8]. Access to laparoscopic simulation is the standard for residency programs in Canada, and it requires a significant investment of equipment and supplies [6, 9, 10]. However, studies of residents' independent use outside of the formal curricula illustrate that uptake of these resources is highly variable [10–14]. Several studies have identified motivation as integral to resident skill acquisition, [8, 14, 15] and some have identified the role that negative external motivators such as lack of time and competing responsibilities play in simulation use [14, 15]. However, to our knowledge, theories of motivation have not been applied to the surgical resident context to determine what motivates residents to undertake independent practice of laparoscopic simulation. With the advent of CBME, understanding resident motivation will be increasingly important to facilitate effective curriculum development.

Our study of resident motivation to use laparoscopic simulation resources is guided by Expectancy Value Theory (EVT), a prominent psychological theory of motivation [16, 17]. From the perspective of EVT, achievement-related choices—such as the decision to voluntarily practice laparoscopic skills in a simulation lab—are understood as a product of an individual's expectancy for success (e.g., Will I be successful if I try?) and subjective task values (e.g., Do I care?). Subjective task values include both positive task values, which contribute to the perceived desirability of task engagement, and negative values, or costs, which make the task seem less desirable. Positive subjective task values include *intrinsic value*, or inherent enjoyment of task engagement; *utility value*, or how the task is perceived to support short- and long-term goals; and *attainment value*, or the personal importance of task engagement. *Costs* refer to what an individual gives up or suffers as a result of engaging in a task, such as effort expenditure, foregone opportunities, and emotional costs [18, 19]. A large body of research in psychology and education has demonstrated that expectancies and subjective task values are associated with achievement, performance, and choice [20]. Therefore, EVT provides a useful

theoretical framework for understanding how various self and task perceptions relate to residents' independent engagement in laparoscopic simulation training. We hypothesize that resident motivation in the laparoscopic simulation context will reveal similar associations to other educational contexts. We feel that EVT is specifically relevant to this context given that this theory explores the connection between extrinsic motivation, cost of engaging in a task, and perception of the utility value of the task as it relates to the end goal. In a busy, mastery focused setting such as residency, we wished to capture these factors comprehensively to understand how they influence resident motivation and felt that EVT represented the most appropriate framework through which to explore this.

The primary objective of this study was to determine what motivates Ob/Gyn residents to engage in laparoscopic simulation use outside of the formal curriculum. To evaluate this, we administered a questionnaire grounded in Expectancy Value Theory that had previously been piloted in a prior study to Ob/Gyn residents in Canadian residency programs [21].

## Materials and methods

Canadian Ob/Gyn residents from English-language residency training programs in their second to fifth year of training and engaged in clinical duties during January-March 2018 were invited to participate. Residents not engaged in clinical duties (e.g. on maternity leave, leave of absence, research programs) were excluded. First year residents were also excluded, as their curriculum in Canadian Ob/Gyn residency programs includes a variable, broad-scope foundation year with limited time on gynecology services, and therefore, likely have limited exposure to laparoscopic simulation and minimally invasive gynecologic surgery. French language residency training programs were not invited to participate, as we had only collected validity evidence for the questionnaire in English [21].

A request was sent to the Program Assistants at all Canadian residency programs to distribute the questionnaire to their residents. Reminder emails were sent out two and four weeks after the initial invitation to further facilitate recruitment. The questionnaire was administered using Opinio Software (Version 7.9.1, ObjectPlant$^{©}$ 1998–2020) and can be found released under a CC-BY 4.0 license (S1 Appendix).

Based on 2018 Canadian Resident Matching Service (CaRMS) data, there were approximately 300 residents in year two to five of training at the time of our study; however, this includes residents who were not engaged in clinical training at the time. Therefore, Program Assistants of the 13 eligible, English language Ob/Gyn programs were asked via email to identify the number of residents engaged in clinical duties in their program to determine the number of eligible participants. Four programs declined to participate or did not respond. Responses indicated 205 residents from 9 programs were eligible to participate. All questionnaire participants were offered entry in a chance to win one of three gift cards to recognize their contribution of time to the study. We obtained approval for this study from our institutional Research Ethics Board (IWK REB Project No. 1022965 on December 13, 2017).

The questionnaire included measures of EVT and several introductory questions to collect demographic data and career aspirations adapted from a previous study of simulation use [14, 22, 23]. Measures of EVT were grouped by component of motivation and presented as 5-point Likert scales. The questionnaire was branched such that residents responded to questions based on exposure to independent practice of simulation and minimally invasive surgery. Table 1 presents EVT scales, example items, and Cronbach's alpha estimates. All scales showed acceptable internal consistency, and therefore means for scales were calculated, with higher values indicating greater endorsement of the construct by the respondent. In addition, participants reported the number of hours in the past 12 months they had spent using simulation

**Table 1. Questionnaire scales.**

| Component of Motivation | Description | Example Item | | | | | Number of Scale Items | Cronbach alpha |
|---|---|---|---|---|---|---|---|---|
| Self-Efficacy for MIS | Perceived ability to independently execute MIS | How good at minimally invasive surgery are you? | | | | | 9 | .949 |
| | | Not at all good | Very good | | | | | |
| | | 1 | 2 | 3 | 4 | 5 | | |
| Self-Efficacy for learning MIS | Perceived ability to learn MIS skills | I am confident that I will be able to learn the skills required to become proficient at minimally invasive surgery | | | | | 7 | .937 |
| | | Not at all confident | Very confident | | | | | |
| | | 1 | 2 | 3 | 4 | 5 | | |
| Perceived Difficulty of MIS | Perceived difficulty of MIS | In general, how hard is performing minimally invasive surgery for you? | | | | | 3 | .814 |
| | | Very easy | Very hard | | | | | |
| | | 1 | 2 | 3 | 4 | 5 | | |
| Intrinsic Interest Value in MIS | Enjoyment derived from performing MIS | In general, I find performing laparoscopic simulation exercises | | | | | 4 | .922 |
| | | Very boring | Very interesting | | | | | |
| | | 1 | 2 | 3 | 4 | 5 | | |
| Intrinsic Interest Value in Simulation Use | Enjoyment derived from simulated practice | How satisfying do you find laparoscopic simulation? | | | | | 4 | .967 |
| | | Not very satisfying | Very satisfying | | | | | |
| | | 1 | 2 | 3 | 4 | 5 | | |
| Attainment Value for MIS | Self-defined importance of succeeding in MIS | How worthwhile is it to master minimally invasive surgery skills to you? | | | | | 5 | .896 |
| | | Not worthwhile | Very worthwhile | | | | | |
| | | 1 | 2 | 3 | 4 | 5 | | |
| Extrinsic Utility Value of MIS | Perceived usefulness of MIS to future goals | How useful is learning minimally invasive surgery to your career goals? | | | | | 5 | .849 |
| | | Not useful | Very useful | | | | | |
| | | 1 | 2 | 3 | 4 | 5 | | |
| | Perceived usefulness of simulated practice for MIS skill development | How transferable are laparoscopic simulation skills to the operating room? | | | | | 5 | .907 |
| | | Not at all transferrable | Very transferrable | | | | | |
| | | 1 | 2 | 3 | 4 | 5 | | |
| Task Effort Cost for Simulation Use | Amount of effort required to engage in simulated practice | Performing laparoscopic simulation exercises takes up too much time | | | | | 5 | .893 |
| | | Strongly disagree | Strongly agree | | | | | |
| | | 1 | 2 | 3 | 4 | 5 | | |
| Other Task Effort Cost for Simulation Use | Amount of time and work put towards other tasks that may impeded simulated practice | Because of the other demands on my time, I don't have enough time to perform laparoscopic simulation exercises | | | | | 4 | .969 |
| | | Strongly disagree | Strongly agree | | | | | |
| | | 1 | 2 | 3 | 4 | 5 | | |
| Loss of Valued Alternatives for Simulation Use | Amount of time taken away by simulated practice from the pursuit of other activities | Performing laparoscopic simulation exercises causes me to miss out on too many things that I care about | | | | | 4 | .896 |
| | | Strongly disagree | Strongly agree | | | | | |
| | | 1 | 2 | 3 | 4 | 5 | | |
| Emotional Cost of MIS | Prominence of negative psychological states that results from MIS, such as anxiety, stress, and fatigue | Performing laparoscopic simulation exercises is too frustrating | | | | | 6 | .938 |
| | | Strongly disagree | Strongly agree1 | | | | | |
| | | 1 | 2 | 3 | 4 | 5 | | |
| Emotional Cost of Simulation Use | Prominence of negative psychological states that results from simulated practice, such as anxiety, stress, and fatigue | I worry too much about performing laparoscopic simulation exercises | | | | | 6 | .919 |
| | | Strongly disagree | Strongly agree | | | | | |
| | | 1 | 2 | 3 | 4 | 5 | | |

MIS refers to minimally invasive surgery

resources and had spent conducing minimally invasive surgery in the operating room. Participants also reported reasons for accessing simulation use, when they accessed simulation resources, barriers to simulation access, percentage of simulation use with supervision, and level of concern about developing bad habits during unsupervised simulation time. The questionnaire is available for review in Supplemental Digital Appendix A.

Proportions were used to conduct content analysis of open-ended responses about barriers to use of laparoscopic simulation. Data from response questionnaires were organized and tallied using R version 3.2.3 (2015-12-10, libraries dplyr v0.7.8, readxl v1.2.0, tidyr v0.8.2). We used one-tailed Spearman correlation tests to examine correlations between EVT scales, hours spent performing laparoscopic simulation, and hours spent performing minimally invasive surgery. Spearman correlations are appropriate for ordinal survey data when linear relations are not assumed; single-tailed tests are appropriate given the extensive literature on the directional relations of EVT constructs and choice behavior [20]. Subgroup analyses using Analysis of Variance (ANOVA) tests were completed to measure differences in EVT scales, simulation use, and minimally invasive surgery experience between junior (Postgraduate Year (PGY) 2–3) and senior (PGY 4–5) residents. SPSS (IBM Corp. Released 2017. IBM SPSS Statistics for Macintosh, Version 25.0. Armonk, NY: IBM Corp.) was used to complete these analyses.

## Results

The questionnaire was distributed to 205 residents. 55 residents started the questionnaire (27% response rate). 10 residents did not complete the questionnaire, and 1 resident who identified as a PGY-1 was excluded. Therefore, 44 residents from 9 residency programs completed the questionnaire and were included in our analysis.

There was a near equal number of junior (PGY 2–3) and senior (PGY 4–5) resident respondents (n = 20, 46% vs. n = 24, 55%). The majority of respondents were female (n = 37, 84%), had previous experience with laparoscopic simulation (n = 43, 98%), and planned to incorporate laparoscopy (n = 40, 91%) into their practice after residency.

### Reasons for accessing and barriers to use of laparoscopic simulation

Residents' self-reported reasons for simulation use are presented in Fig 1. Skill development (n = 38, 88%), free time (n = 36, 84%), and interest in laparoscopic simulation (n = 29, 67%) were the cited as most important reasons for simulation use. Additionally, over half (n = 27, 63%) cited proximity of the simulation lab as important or very important.

More than half of residents accessed simulation outside of work hours or while on vacation (n = 28, 65%). Residents most commonly cited lack of time as a barrier to simulation use (n = 19, 43%). Residents also cited lack of supervised practice or direction regarding simulation exercises (n = 12, 27%) and inconvenient location (n = 5, 11%) as barriers. Most residents used laparoscopic simulation outside of the formal curriculum for between 10 or fewer hours in the last 12 months (n = 37, 84%). For 89 percent of residents (n = 38), less than half of this time was directed practice overseen by staff, fellows, or senior residents. When asked about developing bad habits during unsupervised practice, 61 percent of residents (n = 27) reported being somewhat to very concerned about this occurring.

### Correlations with simulation use and surgical experience

Correlation between expectancy-value constructs and hours of voluntary simulation use and hours of surgical experience are presented in Table 2 and Fig 2. As expected, self-efficacy beliefs and positive task values (e.g., interest, utility, attainment) tended to positively correlate with simulation use and surgical experience, and costs tended to correlate negatively. The

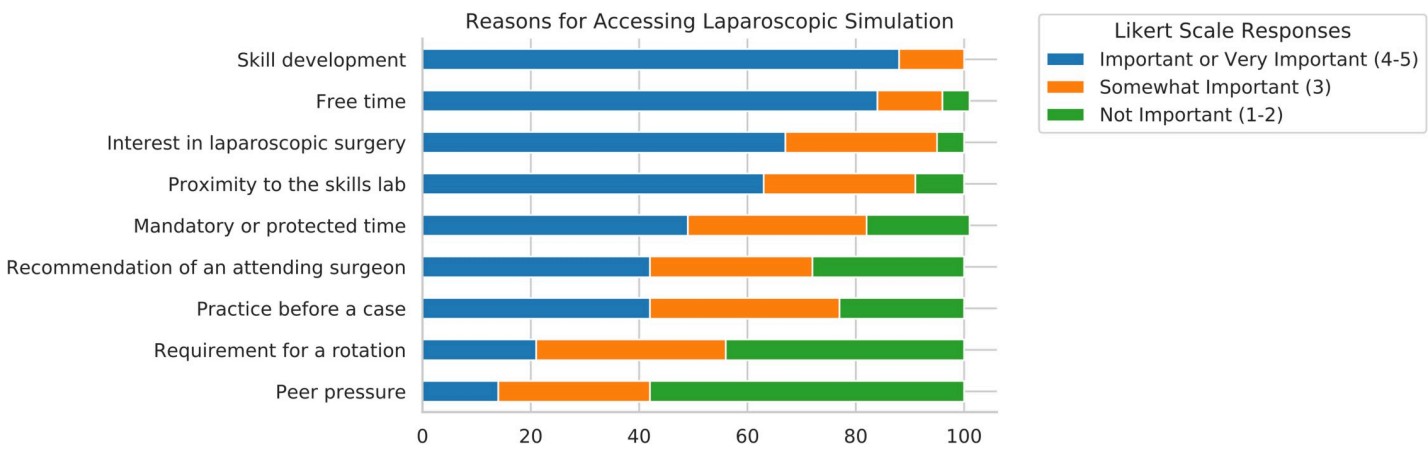

**Fig 1. Reasons for accessing laparoscopic simulation.**

number of hours of simulation use was significantly positively correlated with perceived utility of the simulation for laparoscopic skill development ($rho$ = .310), indicating that residents who perceived that simulated practice translated into applicable surgical skills used the simulation more often.

Hours of experience with minimally invasive surgery was significantly positively correlated with intrinsic interest in laparoscopic surgery, self-efficacy for learning laparoscopic skills ($rho$ = .590) and self-efficacy for doing laparoscopic surgery ($rho$ = .440). Residents who spent more time conducting laparoscopic surgery in the operating room reported enjoying laparoscopy more, and felt more self-efficacious for learning laparoscopic skills and doing laparoscopic surgery. Hours of experience with minimally invasive surgery was also significantly

**Table 2. Spearman's rho correlations between expectancy-value constructs, simulation use, and surgical experience.**

| EVT construct | Hours of Simulation Use | Hours of Minimally Invasive Surgery |
|---|---|---|
| Intrinsic interest in simulation exercises | 0.195 | 0.095 |
| Intrinsic interest in laparoscopic surgery | 0.161 | 0.273* |
| Attainment value of laparoscopic skill development | 0.029 | -0.022 |
| Utility of laparoscopic skills for career | -0.013 | 0.045 |
| Utility of simulation for laparoscopic skill development | 0.310* | 0.128 |
| MIS ability beliefs | 0.059 | 0.254 |
| Self-efficacy for learning laparoscopic skills | 0.173 | 0.590** |
| Self-efficacy for laparoscopic surgery | 0.135 | 0.440** |
| Difficulty of laparoscopic surgery | -0.134 | -0.189 |
| Task effort cost | -0.093 | 0.050 |
| Other task effort cost | -0.246 | -0.158 |
| Loss of valued alternatives | -0.002 | -0.252* |
| Emotional costs of simulation exercises | 0.209 | -0.271* |
| Emotional costs of laparoscopic surgery | 0.118 | -0.341* |

*$p < .05$

**$p < .01$

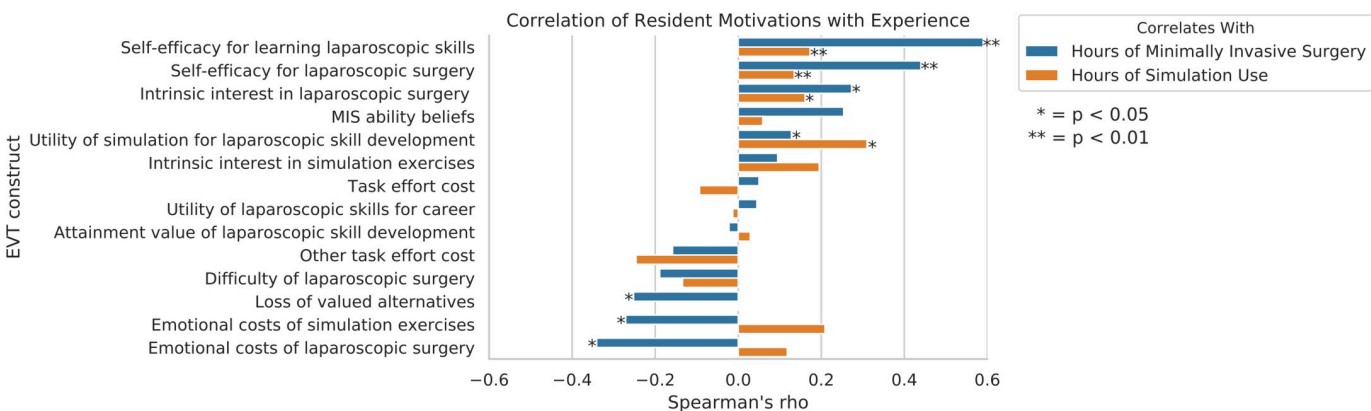

**Fig 2. Spearman's rho correlations between expectancy-value constructs, simulation use, and surgical experience.**

negatively associated with perceived costs, including perceived loss of valued alternatives, emotional costs of simulation exercises, and emotional costs of laparoscopic surgery. That is, residents who spent more time conducting minimally invasive surgery perceived using the simulation less negatively in terms of lost opportunities and negative emotions associated with simulated practice and with conducting laparoscopy in the operating room. Together, these relations suggest experience in the operating room may translate into stronger interest in and self-efficacy beliefs for learning laparoscopy and lowered perceived costs of using simulation resources.

## Differences between junior and senior residents

We conducted a series of ANOVA tests to measure the extent to which Junior (PGY2-3) and Senior (PGY4-5) residents differed in their experience with and motivational beliefs about laparoscopic simulation resources and minimally invasive surgery. Results (presented in Table 3) indicated that, compared to Senior residents, Juniors residents reported significantly more intrinsic interest in laparoscopic surgery while also perceiving laparoscopic surgery as significantly more difficult and involving significantly greater emotional costs. Senior residents reported significantly higher self-efficacy beliefs for both learning laparoscopic skills and for conducting laparoscopic surgery compared to Juniors. All significant results had large effect sizes, according to methods and standards put forth by Cohen [24]. The two groups did not significantly differ in the number of hours spent performing laparoscopic simulation, their intrinsic interest in simulation exercises, utility and attainment beliefs, or costs other than the exception noted above.

## Discussion

This survey of Ob/Gyn residents in English language training programs across Canada used a questionnaire grounded in EVT to measure motivation to access laparoscopic skills training outside of the formal curriculum.

Our findings suggest that most respondents used laparoscopic simulation outside of the formal curriculum for 10 or fewer hours in the past 12 months. This is consistent with the voluntary uptake reported in the study by Chang *et al.* in which 31 percent of residents used the simulation trainer at least once in the 3 months prior to the study and 14 percent of residents completed the training curriculum [12]. Together, these studies indicate poor use of an expensive resource [6, 9, 10]. Further, simulation provides an important opportunity for repetition

**Table 3. Differences between junior and senior residents in motivations for and experience with laparoscopic simulation and minimally invasive surgery.**

| | Junior Residents Mean Difference (+/- SD) | Senior Residents Mean Difference (+/- SD) | F | P-value | Significance | Partial Eta Squared |
|---|---|---|---|---|---|---|
| Hours in simulation in past 12 months | 7.00 +/- 6.50 | 7.55 +/- 6.86 | 0.07 | .794 | | 0.00 |
| Hours in laparoscopic surgery in past 12 months | 9.88 +/- 8.78 | 15.40 +/- 10.25 | 1.48 | .231 | | 0.04 |
| Intrinsic interest in simulation exercises | 3.70 +/- 0.90 | 3.21 +/- 0.93 | 2.75 | .106 | | 0.07 |
| Intrinsic interest in laparoscopic surgery | 4.23 +/- 0.73 | 4.71 +/- 0.46 | 7.37 | .010 | * | 0.16 |
| Utility of laparoscopic skills for career | 4.75 +/- 0.38 | 4.71 +/- 0.64 | 0.16 | .687 | | 0.00 |
| Utility of simulation for laparoscopic skill development | 3.73 +/- 0.90 | 3.55 +/- 0.74 | 0.32 | .577 | | 0.01 |
| Attainment value of laparoscopic skill development | 4.77 +/- 0.39 | 4.78 +/- 0.42 | 0.01 | .921 | | 0.00 |
| Self-efficacy for learning laparoscopic skills | 3.79 +/- 0.65 | 4.42 +/- 0.46 | 10.06 | .003 | ** | 0.21 |
| Self-efficacy for laparoscopic surgery | 2.59 +/- 0.86 | 4.09 +/- 0.51 | 39.29 | < .001 | ** | 0.51 |
| Difficulty of laparoscopic surgery | 3.27 +/- 0.69 | 2.48 +/- 0.68 | 13.20 | .001 | ** | 0.26 |
| Task effort cost | 2.50 +/- 0.54 | 2.40 +/- 0.84 | 0.01 | .947 | | 0.00 |
| Other effort cost | 3.47 +/- 1.00 | 3.41 +/- 1.18 | 0.05 | .817 | | 0.00 |
| Loss of valued alternatives | 2.82 +/- 0.95 | 2.39 +/- 0.78 | 1.77 | .191 | | 0.05 |
| Emotional costs of simulation exercises | 1.94 +/- 0.75 | 1.54 +/- 0.60 | 2.28 | .139 | | 0.06 |
| Emotional costs of laparoscopic surgery | 2.57 +/- 0.75 | 1.70 +/- 0.59 | 16.86 | < .001 | ** | 0.31 |

$^*p < .05$

$^{**}p < .01$

and practice prior to clinical exposure in the operating room setting, an opportunity of particular importance to Ob/Gyn residents, as they have fewer surgical rotations during their training compared to residents in other surgical specialties [5]. Voluntary uptake of laparoscopic simulation use is limited, and this may warrant novel strategies to promote use and consideration of formal integration into curricula.

Previous studies have suggested that residents overwhelmingly believe that laparoscopic simulation leads to skill development that is transferrable to the operating room [14, 22]. In our pilot study [21], residents in a focus group expressed some doubt about the transferability and utility of independent practice of laparoscopic skills. In the current study, we measured this concept directly with multiple survey items and found that 63 percent of residents in this study reported concerns about developing bad habits during unsupervised simulation use. This concern is further reflected by the fact that 27 percent of respondents identified lack of supervision or direction as a barrier to use. Critically, belief in the utility value of simulation was significantly positively correlated with simulation use. It is challenging to ascertain whether this finding represents a shift in thinking within the resident population from the time when the previous studies were conducted, an example of inter-surgical specialty variation in attitudes, or a subset of resident views that were captured in this study. In their review of the teaching and learning theories that underpin CBME, Caccia *et al.* describe a cyclic process of direct observation, feedback, and directed teaching as integral to the CBME framework [25]. Given this and respondents' perceptions of independent engagement in laparoscopic simulation, perhaps curricula renewal should include more supervised practice to facilitate effective acquisition of laparoscopic skills and resident engagement in this resource. At minimum, curricula might educate residents about the evidence for the transferability of skills from simulated practice to the operating room. How best to change resident perspectives in this regard represents an area for future, dedicated exploration.

Self-efficacy can be thought of as the formal psychological construct that most closely captures confidence about a particular task [26]. We found that residents who spent more time performing minimally invasive surgery had greater self-efficacy beliefs about both their ability to master laparoscopic simulation exercises and become competent in minimally invasive surgery and that senior residents, compared to junior residents, had greater self-efficacy beliefs in both domains. Although this is a cross-sectional study without longitudinal examination of the development of beliefs, this may suggest that self-efficacy beliefs strengthen with exposure to minimally invasive surgery. Interestingly, self-efficacy beliefs were not significantly correlated with voluntary use of simulation resources, suggesting no clear link in the current study between simulation practice and self-efficacy belief for learning and performing minimally invasive surgery. As stronger self-efficacy beliefs are shown to positively correlate with personal accomplishment and general psychological well-being and negatively correlate with emotional exhaustion in surgical residents [27], the facilitation of self-efficacy by surgical educators through feedback, coaching, and directed educational opportunities is important to both the well-being of surgical residents and their achievement. Further, although research exploring self-efficacy beliefs in medical education is growing [28], our review of the literature suggests our study is among the first to explore self-efficacy beliefs for minimally invasive surgery during surgical residency training. As we move towards CBME, there may be a novel opportunity to explore this trajectory in a longitudinal fashion.

In comparing junior and senior residents, junior residents experience greater emotional costs associated with performing laparoscopic surgery. This finding may be important to educators in that they may wish to consider the greater emotional toll that performing laparoscopic surgery poses for junior residents in an effort to ensure that there is focus on a safe, effective learning environment for both the development of competence and self-efficacy.

While resident motivation has frequently been cited as an important area for research in surgical education [8, 14, 15], to our knowledge, existing literature has yet to leverage psychological theories of motivation to guide investigations. A major strength of this study is that it examines resident motivation through the theoretical lens of EVT, and this represents a novel application of this theory to the surgical education field and to surgical skill acquisition. Additionally, this is a National survey of Ob/Gyn residents and represents perspectives from 9 residency programs, which positively influences its generalizability and applicability to surgical educators.

There are several limitations to this study. As with any questionnaire-based study, this study is subject to selection bias, which limits its generalizability. In addition, the response rate was lower than expected, which increased the risk of response bias. Unfortunately, surveys of medical residents seem to frequently experience this difficulty of low response rates (e.g. 8.3% [29], 22% [30], 27% [31], 27.9% [32]). Based on research investigating this problem, the most likely reason for our observed response rate was the time burden of completing the study [33]. As this could bias our responses towards residents with more available time, we might be artificially suppressing certain EVT constructs such as the "Task Effort Cost for Simulation Use". An approach that might lead to higher response rates would be to administer the survey in conjunction with the National Practice Examination, which is completed annually. Further, although our sample size is reflective of the small number of Ob/Gyn residents in Canada, we lacked the statistical power to detect small effect sizes. Despite these constraints, our analyses were able to identify prominent ways in which motivational beliefs relate to training behaviors. We also relied on self-reports of amount of simulation use and Minimally Invasive Surgery (MIS) exposure, which are not as precise as objective behavioral measures. Finally, while cross-sectional analyses examining differences between junior and senior residents provided a window into attitudes towards simulation and MIS during different phases of residency,

longitudinal studies will be required to explore the developmental trajectories of belief and values. Despite these limitations the novel exploration of motivation within the context of surgical training offers actionable insights to support evidence-based curriculum renewal.

## Conclusion

This national study of Ob/Gyn residents explored what motivates residents to engage in laparoscopic simulation outside of the formal curriculum. Guided by an expectancy-value theoretical framework, we found that perceived utility of simulation for skill development was positively correlated with simulation use and self-efficacy beliefs were higher for senior compared to junior residents. Respondents expressed doubt about the utility of independent practice for skill acquisition, and, overall voluntary use of laparoscopic simulation was limited. These findings represent important areas of focus for surgical educators in curricula renewal and interventional study design to optimize the use of laparoscopic simulation to facilitate laparoscopic skill acquisition and ensure resident engagement in this important resource.

## Supporting information

**S1 Appendix. Canadian survey of Ob/Gyn resident' motivation to access lapropscopic simulation training.**
(DOCX)

**S1 Table. Aggregated survey responses for reasons for accessing laparoscopic simulation.**
(DOCX)

## Author Contributions

**Conceptualization:** Jocelyn Stairs, Bradley W. Bergey, Stephanie Scott.

**Data curation:** Jocelyn Stairs.

**Formal analysis:** Bradley W. Bergey, Finlay Maguire.

**Funding acquisition:** Jocelyn Stairs.

**Investigation:** Jocelyn Stairs.

**Methodology:** Jocelyn Stairs, Bradley W. Bergey.

**Project administration:** Jocelyn Stairs.

**Supervision:** Stephanie Scott.

**Visualization:** Finlay Maguire.

**Writing – original draft:** Jocelyn Stairs.

**Writing – review & editing:** Jocelyn Stairs, Bradley W. Bergey, Finlay Maguire, Stephanie Scott.

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
