## [Decision Letter · Decision Letter 0]

30 Jan 2020

PONE-D-19-34982

Motivation to Access Laparoscopic Skills Training: Results of a Canadian Survey of Obstetrics and Gynecology Residents

PLOS ONE

Dear Dr Stairs,

Thank you for submitting your manuscript to PLOS ONE. After careful consideration, we feel that it has merit but does not fully meet PLOS ONE’s publication criteria as it currently stands. Therefore, we invite you to submit a revised version of the manuscript that addresses the points raised during the review process.

Please respond to the comments below from the reviewers in your updated manuscript. We look forward to your revision and the opportunity to publish this article.

We would appreciate receiving your revised manuscript by Mar 15 2020 11:59PM. To enhance the reproducibility of your results, we recommend that if applicable you deposit your laboratory protocols in protocols.io, where a protocol can be assigned its own identifier (DOI) such that it can be cited independently in the future. For instructions see: http://journals.plos.org/plosone/s/submission-guidelines#loc-laboratory-protocols

We look forward to receiving your revised manuscript.

Kind regards,

Amy Michelle DeBaets, PhD

Academic Editor

PLOS ONE

Journal Requirements:

2. Please clarify in your Methods section whether the questionnaire is published under a CC-BY license, or whether you obtained permission from the publisher to reproduce the questionnaire in this manuscript.

Please explain any copyright or restrictions on this questionnaire.

4. Your ethics statement must appear in the Methods section of your manuscript. If your ethics statement is written in any section besides the Methods, please move it to the Methods section and delete it from any other section. Please also ensure that your ethics statement is included in your manuscript, as the ethics section of your online submission will not be published alongside your manuscript.

5. Please include captions for your Supporting Information files at the end of your manuscript, and update any in-text citations to match accordingly. Please see our Supporting Information guidelines for more information: http://journals.plos.org/plosone/s/supporting-information

Reviewers' comments:

Reviewer's Responses to Questions

**Comments to the Author**

1. Is the manuscript technically sound, and do the data support the conclusions?

Reviewer #1: Yes

Reviewer #2: Partly

2. Has the statistical analysis been performed appropriately and rigorously? 

Reviewer #1: Yes

Reviewer #2: I Don't Know

3. Have the authors made all data underlying the findings in their manuscript fully available?

Reviewer #1: Yes

Reviewer #2: Yes

4. Is the manuscript presented in an intelligible fashion and written in standard English?

Reviewer #1: Yes

Reviewer #2: Yes

5. Review Comments to the Author

Reviewer #1: In this work, authors present a very important aspect of simulation, that is, the motivation to participate in simulation sessions, also independently. The work is well structured and demonstrates that motivation is the real impetus to learning beyond the curricula.

Indeed, belief in the utility value of simulation was significantly positively correlated with simulation use.

This work introduces new elements in the evaluation of a simulation program in trainees and I turn on some interest in those who have dedicated themselves in this type of research.

Reviewer #2: Simulation is currently a fundamental component in surgical training. The article highlights that trainees use this learning method very little and try to identify the causes. However, despite the topic of interest, I believe there are critical issues in the development of the survey. I think that:

- the number of subjects enrolled is too small to represent reality;

- the number of subjects who did not answer the questionnaire is high: how do we interpret this data? Why didn't they answer?

I consider the purposes of the study of interest, above all to understand the problem and find solutions. However, it is necessary to broaden the enrolled subjects and find more specific purposes for the study. The conclusions reached are useful but not decisive.

6. PLOS authors have the option to publish the peer review history of their article (what does this mean?). If published, this will include your full peer review and any attached files.

Reviewer #1: Yes: Paolo Mannella

Reviewer #2: No

---

## [Author Response · Author response to Decision Letter 0]

2 Mar 2020

Journal Requirements:

 Author information, title formatting, citations, tables, and abbreviations have been updated in line with PLOS ONE’s style requirements. 

2. Please clarify in your Methods section whether the questionnaire is published under a CC-BY license, or whether you obtained permission from the publisher to reproduce the questionnaire in this manuscript. Please explain any copyright or restrictions on this questionnaire.

We have added the following text to the Materials and methods section of the manuscript: 

“The questionnaire was administered using Opinio Software (Version 7.9.1, ObjectPlant© 1998-2020) and the full question set used can be found under a CC-BY 4.0 license as Supporting Information S1 Appendix”

The small number of residents in each year of training and the small number of Canadian training sites makes the potential for identification possible. Given the potential implications of residents critiquing their own institution’s teaching environment, even if they were only identified in aggregate by site, our Institutional Ethics Review Board considered the raw responses as sensitive. 

They can be contacted here: 

IWK REB Project No. 1022965 on December 13, 2017

IWK Health Centre Research Services

5850/5980 University Ave

PO Box 9700

Halifax, Nova Scotia, Canada

B3K 6R8

research@iwk.nshealth.ca

We support PLOS ONE’s dedication to reproducibility and replicability in science, as explained above we are unable to release the raw responses under the terms of our ethics approval, however all aggregated responses required to reproduce the figures in the manuscript are available in Table 2 and Supporting Materials S1 Table.

4. Your ethics statement must appear in the Methods section of your manuscript. If your ethics statement is written in any section besides the Methods, please move it to the Methods section and delete it from any other section. Please also ensure that your ethics statement is included in your manuscript, as the ethics section of your online submission will not be published alongside your manuscript.

The separate ethics section has been removed and the Methods section been updated to contain the following:

“We obtained approval for this study from our institutional Research Ethics Board (IWK REB Project No. 1022965 on December 13, 2017).”

5. Please include captions for your Supporting Information files at the end of your manuscript, and update any in-text citations to match accordingly. Please see our Supporting Information guidelines for more information: http://journals.plos.org/plosone/s/supporting-information

We have updated the manuscript to include these in the main text.

Reviewers' comments:

Reviewer's Responses to Questions

Comments to the Author

1. Is the manuscript technically sound, and do the data support the conclusions? 

Reviewer #1: Yes

Reviewer #2: Partly

2. Has the statistical analysis been performed appropriately and rigorously?

Reviewer #1: Yes

Reviewer #2: I Don't Know

3. Have the authors made all data underlying the findings in their manuscript fully available?

Reviewer #1: Yes

Reviewer #2: Yes

4. Is the manuscript presented in an intelligible fashion and written in standard English?

Reviewer #1: Yes

Reviewer #2: Yes

5. Review Comments to the Author

Reviewer #1: In this work, authors present a very important aspect of simulation, that is, the motivation to participate in simulation sessions, also independently. The work is well structured and demonstrates that motivation is the real impetus to learning beyond the curricula.

Indeed, belief in the utility value of simulation was significantly positively correlated with simulation use.

This work introduces new elements in the evaluation of a simulation program in trainees and I turn on some interest in those who have dedicated themselves in this type of research.

We really appreciate the kind comments and interest of the reviewer. We agree that the Expectancy Value Theory approach has a lot to offer in medical education.

Reviewer #2: Simulation is currently a fundamental component in surgical training. The article highlights that trainees use this learning method very little and try to identify the causes. However, despite the topic of interest, I believe there are critical issues in the development of the survey. I think that:

- the number of subjects enrolled is too small to represent reality;

We are grateful for the reviewers consideration of our work and fully agree that the response rate is the largest caveat when interpreting our findings. Unfortunately, these response rates are highly typical of this population:

- 2018 National Survey of Resident Doctors in Canada received an 8.3% response rate

- “A survey of Canadian general surgery residents’ interest in international surgery” by Barton et. al. published in the CMAJ in 2008 received a 27% response rate 

- “A cross-sectional online evaluation of burnout risk factors among general surgical residents in Canada” by Adams et. al. received a 22% response rate.

To emphasise this limitation and hopefully promote analysis of the difficulty in studying medical residents, we have added the following to the discussion section of the manuscript:

“In addition, the response rate was lower than expected, which increased the risk of sampling bias. Unfortunately, surveys of medical residents seem to frequently experience this difficulty with response rates below 33% (e.g. 8.3% [29], 22% [30], 27% [31]).” 

- the number of subjects who did not answer the questionnaire is high: how do we interpret this data? Why didn't they answer?

We agree that the question of why physicians and residents have typically low response rates to surveys is a very interesting question. Indeed, this is an open research question in medical research (e.g. “Exploring physician specialist response rates to web-based surveys.” by Cunningham et. al. published in BMC Medical Research Methodologies in 2015). Cunningham found that survey burden and time constraints were the leading reason for non-response (60.3%). While this is not within the scope of this project we would definitely adopt their methodology to attempt to follow up on non-respondents. 

We’ve added to following to the discussion to highlight this issue and propose a putative solution for future studies:

“Based on research investigating this problem the most likely reason for our observed response rate was the time burden of completing the study [33]. As this could bias our responses towards residents with more available time, we might be artificially suppressing certain EVT constructs such as the “Task Effort Cost for Simulation Use.” 

An approach that might lead to higher response rates would be to administer the survey in conjunction with the National Practice Examination, which is completed annually.”

I consider the purposes of the study of interest, above all to understand the problem and find solutions. However, it is necessary to broaden the enrolled subjects and find more specific purposes for the study. The conclusions reached are useful but not decisive.

We are grateful that the reviewer agrees that motivation in surgical education is an interesting and important question. Unfortunately, as we already distributed the survey to every English speaking obstetrics and gynaecology residency program in Canada it is not possible to further broaden enrolment without reaching out to international programs. Due to the differences in culture and curricula this would involve we feel broadening enrolment like this would further obfuscate any useful inferences. Fundamentally, we believe this work forms an important inductive and hypothesis generating contribution to the medical education literature. Hopefully, this work provides the groundwork for generating deductive interventional or control-curriculum approaches to improve use of simulation among surgical residents. 

We’ve further clarified this in the discussion:

“Despite these limitations the novel exploration of motivation within the context of surgical training offers actionable insights to support evidence-based curriculum renewal.” 

And conclusion:

“These findings represent important areas of focus for surgical educators in curricula renewal and interventional study design to optimize the use of laparoscopic simulation to facilitate laparoscopic skill acquisition and ensure resident engagement in this important resource.”

6. PLOS authors have the option to publish the peer review history of their article (what does this mean?). If published, this will include your full peer review and any attached files.

Do you want your identity to be public for this peer review? For information about this choice, including consent withdrawal, please see our Privacy Policy.

Reviewer #1: Yes: Paolo Mannella

Reviewer #2: No

---

## [Editor Report · Decision Letter 1]

12 Mar 2020

Motivation to Access Laparoscopic Skills Training: Results of a Canadian Survey of Obstetrics and Gynecology Residents

PONE-D-19-34982R1

Dear Dr. Stairs,

We are pleased to inform you that your manuscript has been judged scientifically suitable for publication and will be formally accepted for publication once it complies with all outstanding technical requirements.

With kind regards,

Amy Michelle DeBaets, PhD

Academic Editor

PLOS ONE